**Data Availability Statement:** Data cannot be shared publicly because they consist of individual

# Participation in interventions and recommended follow-up for non-attendees in cervical cancer screening -taking the women's own preferred test method into account—A Swedish randomised controlled trial

**Caroline Lilliecreutz**[‡]*, **Hanna Karlsson, Anna-Clara Spetz Holm**[‡]

Department of Obstetrics and Gynaecology and Division of Children's and Women's Health, Department of Biomedical and Clinical Sciences, Faculty of Medicine and Health Sciences, Linköping University, Linköping, Sweden

‡ These authors are joint senior authors on this work.
* caroline.lilliecreutz@regionostergotland.se

## Abstract

### Background

Cervical cancer is a highly preventable disease. To not attend an organized cervical cancer screening program increases the risk for cervical dysplasia and cervical cancer. The aim was to investigate the participation rate in three different intervention groups for non- attendees in the Swedish national program for cervical screening. The participation in the recommended follow up, and the histology found were also examined.

### Method

Population-based randomized control trial. It included 10,614 women that had not participated in the cervical cancer screening programme during the last six years (ages 30–49) and the last eight years (ages 50–64) were randomised 1:1:1(telephone call from a midwife (offering the choice between a visit for a pap smear or an HPV self-sampling test); an HPV self-sampling test only; or the routine procedure with a yearly invitation).

### Results

In the intention to treat analysis the participation rates were 25.5% (N = 803/3146) vs 34.1% (N = 1047/3068) and 7.0% (N = 250/3538) (p<0.001) for telephone, HPV self-test and control groups respectively. In the by protocol analysis including women that answered the phone call the participation rates were 31.7% (N = 565/1784) vs 26.1% (N = 788/3002) and 7.0% (N = 250/3538) (p<0.001) for telephone, HPV self-test and control groups. The corresponding results in the by protocol analysis including women that did not answer the phone call was 19.7% (N = 565/2870) vs 26.1% (N = 788/3002) and 7.0% (N = 250/3538) (p< 0.001). The majority of the women 63,4% (1131/1784) who answered the telephone wanted to participate either by booking a visit for pap smear (38,5%) or to be sent a HPV self-

medical data that could be connected to a specific person. Data are available on request for any researchers to allow replication of results provided all ethical and legal requirements are met. Data can be requested through Forum Östergötland, Linköping University, with contact person Mats Fredrikson Mats mats.fredrikson@liu.se. Registries used for this study include the Swedish National Cervical Cancer Registry and A local database (HPV data). The website of the ethics committee is: www.Etikprovningsmyndigheten.se (contact via registrator@etikprovning.se).

**Funding:** ALF Grants 022-95984 Region Östergötland.

**Competing interests:** The authors have declared that no competing interests exist.

sampling test (24,9%) (p<0.001). Women who chose an HPV self-test were older and gave anxiety/ fear as a reason to decline participation, and they were also less likely to participate in the follow-up if found to be HPV-positive compared to the women who chose a Pap smear. The attendance to the recommended follow-up after abnormality was in total 87%. The non-attendees had a three or eight times higher risk of having a cytology result of HSIL or suspected SCC respectively, in the index sample compared to women screened as recommended (OR 3.3 CI 95% 1.9–5.2, OR 8.6 CI 1.6–30). A total of ten SCC and one adenocarcinoma were found in the histopathology results from the non-attendee group with a study intervention, while there was only one SCC in the non-attendee group without any study intervention (p = 0.02, OR 8.1 CI 95% 1.2–350).

## Conclusions

Our study suggests, according to intention to treat analysis, that the best intervention to get as many non-attendees as possible to participate is to send an HPV self-sampling test together with an invitation letter. Almost 90% of women in the study with an abnormal index sample attended follow-up. This is high enough to indicate that interventions to increase the participation among non-attendees are meaningful.

## Registry

International Standard Randomised Controlled Trial Number (ISRCTN)
  Registration number ISRCTN78719765.

## Introduction

In Sweden, 559 new cases of cervical cancer were identified in 2017, i.e. 11.2 cases per 100 000 women [1]. However, cervical cancer is highly preventable, and many countries have developed organised cervical cancer screening programmes that in recent decades have contributed to a reduced mortality and incidence of cervical cancer [2]. In 2016, the cervical cancer screening programme in Region Östergötland in Sweden included women 23–64 years old, sending them an invitation for a Papanicolaou (Pap) smear at an interval of every third year between the ages of 23–49, and every fifth year at ages 50–64 years. Infection with high risk human papillomavirus (HPV) is a prerequisite for high-grade intraepithelial cervical lesion (HSIL) and cervical cancer and these lesions do not develop without a persisting infection with the virus [3]. There are at least 13 oncogenic types of HPV, including HPV 16 and 18, which account for approximately 70% of all invasive cervical cancer cases worldwide [4]. Within two years after the HPV infection, slightly more than 90% of young women clear the infection spontaneously [5]. One of the most important risk factors for developing cervical cancer is not attending cervical cancer screening programmes regularly. Accepting the regular invitations reduces by 90–95% the risk of developing cervical cancer [6]. A randomised controlled trial from Sweden comparing abnormal Pap smears from attendees with non-attendees found that non-attendees had a fourfold increase in HSIL [7]. Reasons for not attending cervical cancer screening programmes are lack of accessibility to the clinic, mental and physical illness, disability, a feeling of being healthy, sexual abuse, sexual inactivity or dislike of gynaecological examinations. Women from low socioeconomic areas and new residents with a poor understanding of the Swedish language also attend to a lower degree [7–9]. Other co-factors for developing

HSIL and cervical cancer are low socioeconomic status, smoking, having different sexual partners and a personal history of sexually transmitted diseases. Smoking and immunosuppression such as HIV infection facilitate the progression from HPV infection to HSIL and invasive squamous cell carcinomas (SCC) [10]. Earlier interventions to reach non-attendees have been studied including telephone calls from a midwife or invitations to take an HPV self-test outside the clinic. Both inventions increased the participation rate [7, 11]. In another study from Belgium, women that received an HPV self-test participated more often compared to women who ordered the HPV self-test after receiving an invitation letter (18.7% vs 10.6%) [12]. Since the non-attendees are at high risk for developing HSIL and cervical cancer, it is important to approach this group with appropriate interventions other than the regular yearly invitations, and also to see that they attend recommended follow-ups. Therefore, it was indicated that other interventions to reach this group were needed.

We hypothesised that the participation rate would increase if the non-attendees could choose between different test methods presented by a midwife in a telephone call (an HPV self-test or Pap smear in the clinic) or if they received an HPV self-test directly by post for sampling rather than yearly invitations (routine procedure). Therefore the primary aim of this study was to evaluate the participation rate comparing different interventions among the non-attendee women and to describe the attendance to recommended follow-up (if HPV and/or cervical dysplasia were present). The secondary aim was to determine what intervention the non-attendees would choose if, in a telephone call with a midwife, they were given the possibility to choose between an HPV self -test outside the clinic and a visit for a Pap smear. We also hypothesised that we would find more abnormal index samples in non-attendees than in attendees. Our third aim was therefore to describe the cytological diagnoses found in the index Pap smear and compare them with those found in attendees during the same period. The histological diagnosis found in the follow-up will also be described.

## Material and methods

Non-attendees were defined as women who had not participated in the cervical cancer screening programme during the last six years if aged 30–49, and during the last eight years if aged 50–64. This strategy was in accordance with the Swedish age-differentiated screening intervals at that time. The population was defined as non-attendees living in the region of Östergötland, Sweden on March 8th 2016. Data were extracted from the Swedish national cervical cancer screening registry (NKCx) which has almost 100% coverage of invitations. The results of Pap smears and biopsies can be extracted from the personal identification number, which is unique to each individual. The results from the HPV self-tests were extracted from a local registry in the clinic. Data concerning follow-up results were obtained until the 31th of December 2017 when the study period ended. The non-attendees' addresses were obtained from the Swedish Population Register (SPR). For comparison, all women (N = 44 938) from the region of Östergötland that completed the last two previous cervical cancer screening rounds were included in the analysis concerning cytology results. They were defined as the attended group. The data were extracted from the Swedish national cervical cancer screening registry (NKCx).

The non-attendees were randomly, 1:1:1, assigned to one of three different groups: telephone, HPV self-test or control with no intervention except for the yearly invitations (routine procedure). The study groups were computer randomised and adjusted for area code and age by a statistician uninvolved in the study. All three groups received the yearly invitations which were distributed equally over the study period. The first analysed sample, Pap smear or HPV test, in all groups is referred to as the index sample. We considered that the different interventions had an effect if an HPV self-test was ever returned during the study period or a Pap

smear was analysed within six months from the date a study invitation letter was sent. Non-attendees randomised to the telephone group were offered a choice of different sampling options, as presented below. Invitation letters were posted between April 12th 2016 and 31st May 2017.

### HPV self-test group

Free of charge HPV self-tests (Cobas® PCR Media Uni Swab kit) were distributed between April 2016 and May 2017 to non-attendees randomised to the HPV self-test group, accompanied by an invitation letter with information about the aim of the study and instructions on how to correctly collect the sample, together with descriptive illustrations. The invitation letter gave information about recommended follow-up if the HPV self-test was positive. A pre-paid envelope was included in the invitation letter. The participants were instructed first to wash their hands, second to insert the swab at least 5 centimetres into the vagina, and rotate the swab for approximately 30 seconds. They were instructed then to place the swab into a test tube, avoiding contamination with the surrounding environment. The women were asked to carefully check that the personal identification number was correct before affixing the pre-printed labels onto the test tube. If the HPV self-test was not returned within a month a reminder letter was sent. If the HPV self-test returned was positive, the women were booked for a colposcopy at the outpatient clinic. Non-attendees with a negative HPV were informed of the result by letter and told that in the future they would be invited to attend screening according to the regular cervical cancer screening programme.

### Telephone group

Telephone numbers were collected by a medical secretary from different sources such as medical records, local registries and websites listing telephone numbers. Only when a telephone number was found, a study invitation letter with information was possible to send. The letters were sent between April 2016 and February 2017. In total, 24 midwives called non-attendees within a month from the date the invitation letter was sent, offering an appointment with a midwife for a Pap smear at the out-patient clinic, or the option to receive an HPV self-test at the address registered in the SPR, both options free of charge. The telephone call was presented with a number from the hospital if the women could see the sender on the display on their telephones. Three attempts were made to reach the non-attendees. Telephone calls were mainly carried out in the evenings between 5- and 8 pm. All midwives participating in the study underwent education about HPV and the cervical cancer screening programme. Questions about reasons for not participating in the cervical screening programme were also asked and all midwives followed a manual during the telephone call. Information about the association between HPV and cervical cancer was given. Non-attendees who chose an HPV self-test also received study information by a letter and instructions (see above). No reminder letter was sent to this group. The above described procedure concerning positive versus negative HPV results for the women choosing an HPV self-test was also applied with the telephone group. If an appointment for a Pap smear was chosen, the result of the index sample was handled according to the current medical guidelines. Non-attendees who did not choose any of the interventions were recommended to attend the cervical cancer screening programme or to visit the clinic for a Pap smear during drop-in times. If the non-attendee was found to have previously undergone a hysterectomy a special form was sent to an administrator to see if the women could be excluded from the screening program. Non-attendees who said they did not want to participate in the screening programme in the future received a special form by post to complete.

## Sample handling

The HPV self-test was analysed on the Cobas 4800 system [13]. Preparation of the samples was automatically carried out with the Cobas x 480 instrument of extraction. The prepared tests were transferred to the Cobas z 480 instrument of analysis to conduct a real-time polymerase chain reaction (PCR). This method included amplification of targeted sequences and fluorescence, which detects the type of HPV in each PCR-cycle. Eagle Z05 polymerase and specific HPV-primers for HPV type 16, 18, 31, 33, 35, 39, 45, 51, 52, 56, 58, 59, 66 and 68 were used to amplify the target sequence of the HPV- and β-globin-genome. TaqMan probes specifically for HPV 16, HPV 18 and other hrHPV types (HPV type 31, 33, 35, 39, 45, 51, 52, 56, 58, 59, 66 and 68) or β-globin were marked with fluorochrome and a quencher.

The Pap smears were collected with the Surepath liquid-based Pap-test and diagnosed by cytologists in an accredited laboratory [14].

## Follow-up procedure and colposcopy

Non-attendees with a positive HPV self-test or a Pap smear with atypical squamous cells of undetermined significance (ASCUS) or low-grade squamous intraepithelial lesions (LSIL) with positive HPV in the reflex test were booked for examination with colposcopy and possible biopsy at the outpatient clinic. All non-attendees with cytology that could not exclude a high-grade squamous intraepithelial lesion were also investigated with colposcopy and biopsy, but without an HPV test according to the medical guidelines. The results from the colposcopy and biopsy determined future investigations and treatments.

## Statistics

Data were analysed with descriptive statistics. To reach a power of 80% regarding the participation rates in each study arm, 723 non-attendees were needed in each study arm. The rejection of the null-hypothesis was set to 0.05 (two-sided) in all statistical analyses. A student's t-test was used to test differences between quantitative variables. A Pearson's Chi-square test was used for testing differences in proportions between categories. Odds ratios were used as a measure of association, and are presented with exact confidence limits.

The intention to treat analysis included the study population randomized to each group excluding the individuals that had given a Pap smear before the invitation letter was sent (Fig 1).

The by protocol analysis included the study population randomized to each group, excluding the individuals that had given a left Pap smear before the invitation letter was sent or who did not have an address (the HPV self-test group and Telephone group) and did not have phone number (the Telephone group). A comparison was made in regard to whether the women answered the phone call from the midwife or not (the Telephone group) (Fig 1).

All statistical analyses were carried out done with Stata v 15.1, Stata Corp LLC, College Station, TX, USA.

## Ethical approval

This study was approved by the Regional Ethical Review Board in Linköping, Sweden (Dnr 2015/480-31 dated 27/01/2016 and 12/04/2016) which determined that written informed consent by the study participants was not required. The final approval from the Regional Ethical review board concerning the information to the study population, came after the randomization had taken place but before the invitation letters were sent. The consent was giving verbally and/or confirmed by taking a sample (Pap smear or HPV self-test) after the study information

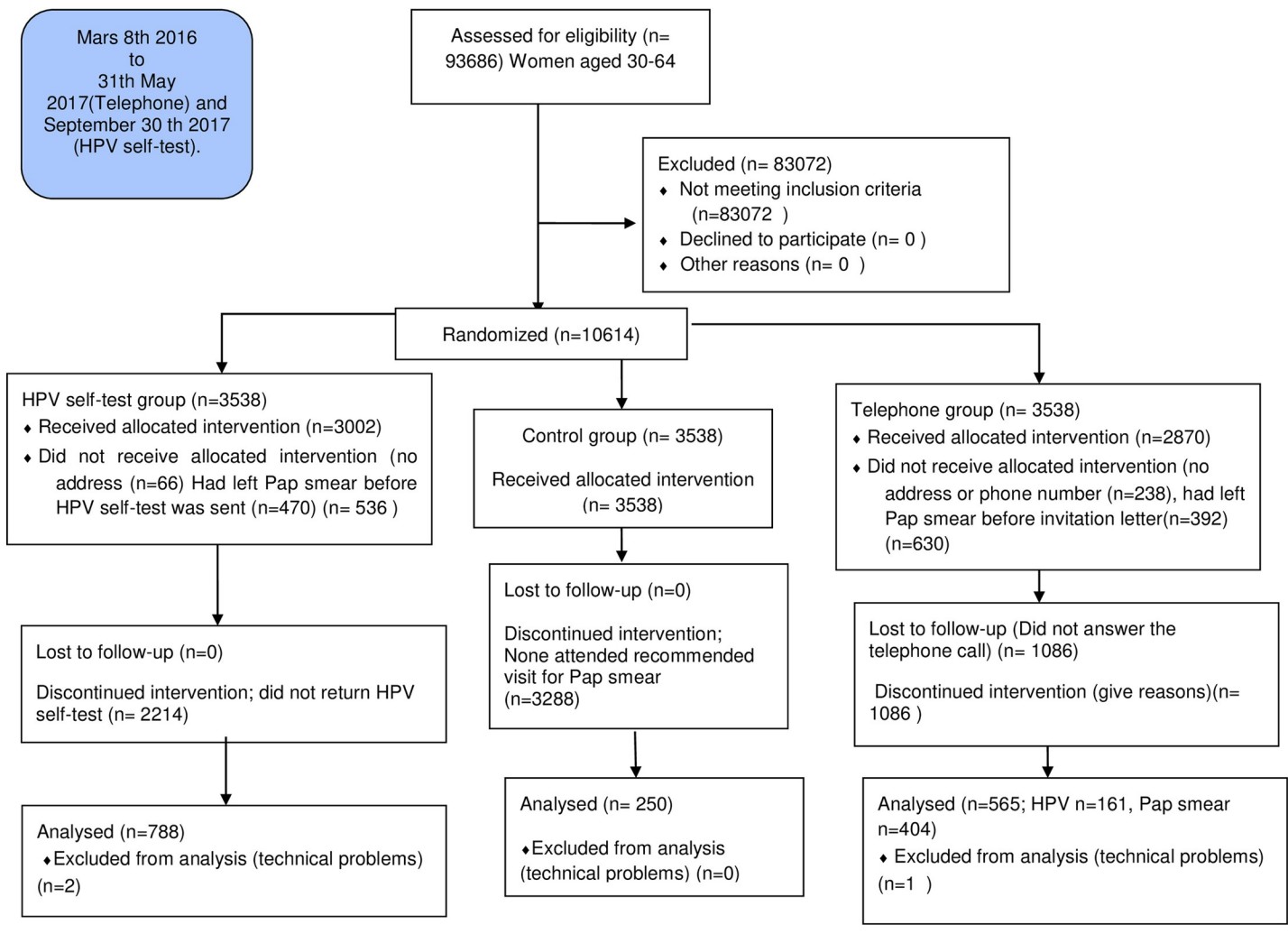

**Fig 1. Study population.**

was given. The attending group of women were extracted anonymously from the Swedish national cervical cancer screening registry for comparison reasons. The trial was not registered in ISRCTN or ClinicalTrials.gov before the enrolment of participants started because this was not recommended by the ethical board.

## Results

The study population is shown in Fig 1. Of 93686 women in the county of Östergötland, aged 30–64 years, 10614 (11.3%) met the criteria for non-attendees in the national cervical cancer screening programme on the 8th of March 2016.

### Participation in the different study groups

*The HPV self-test group* consisted of 3538 women, but since 470 women had provided a Pap smear between the date of randomisation and the date the study invitation letter with the HPV self-test was sent they were excluded. The intention to treat group consisted of 3068 (3002 + 66 without address) women and 788 HPV self-tests were returned (788/3068, 25.7%). In this

**Table 1. Comparison between the intervention groups regarding index sample (Pap smear and /or HPV).**

| Intention to treat[*] | Telephone group N = 3146 N (%) | | | HPV self-test group N = 3068 N (%) | p-value | Control group N = 3538 N (%) | p-value |
|---|---|---|---|---|---|---|---|
| Indexsample[**] | 803 (25.5) | | | 1047 (34.1) | <0.001 | 250 (7.0) | <0.001 |
| **By Protocol**[*] | **Telephone group with answers N = 1784 N (%)** | **Telephonegroup without answers N = 2870 N (%)** | | **HPV self-test group N = 3002 N (%)** | | **Control Group N = 3538 N (%)** | |
| Indexsample[**] | 565 (31.7) | 565 (19.7) | | 788 (26.2) | <0.001 | 250 (7.0) | <0.001 |
| Women's choice of test | Pap SmearN = 686N (%) | HPV N = 445N (%) | | | | | |
| Indexsample[**] | 404 (58.9) | 161 (36.2) | | | | | <0.001 |

* For definition of Intention to treat and By Protocol analysis see statistics section

** Index sample: The first analysed sample, Pap smear or HPV test

group, 259 women provided a Pap smear within six months after the date the invitation letter with the HPV self-test was sent instead of returning the HPV self-test, making 1047 (34.1%) index samples in total. Of the women that received an HPV self-test, 26.2% returned a sample (788/3002 by protocol) (Table 1).

Of the HPV self-tests 36.8% (N = 290/788) were received after a reminder letter was sent (Fig 1). In total, 2504 reminder letters were sent which resulted in 290 additional HPV self-tests (11.6%). The mean time from when the HPV self-test was sent to the completion of analysis was 28 days. Two samples could not be analysed due to technical problems.

*The telephone group* consisted of 3538 women. The intention to treat group consisted of 3146 women since 392 women had provided Pap smears before the study invitation letter was sent. Another 668 were excluded for different reasons. In total, 2870 women received a phone call from a midwife and 62.2% (1784/2870) answered the phone (Fig 1, Table 1).

In the intention to treat group, 803 (803/3146 25.5%) index samples, Pap smears (N = 642) or HPV self-tests (N = 161) were analysed (Table 1). Of the women who answered the phone, 404 provided Pap smears while 238 women provided Pap smears within six months after the study invitation letter was sent despite not answering the phone (Fig 1). Index samples were left by 565 (404 Pap smears and 161 HPV self-tests) women in total, corresponding to 19.7% of the women who received a phone call (565/2870) and 31.7% (565/1784) of the women who answered the telephone (by protocol) (Table 1). One HPV self-test could not be analysed due to technical problems.

The mean age of the women who answered the telephone call was 49.9 years and for those who did not it was 48.7 years (p = 0.005). After one telephone call, 58% (N = 1035) of the women answered and 27% (N = 482) respectively 15% (N = 267) after the second and third calls. The outcome of the phone call is shown in Table 2.

**Table 2. Outcome of the telephone call with the midwife.**

| | N = 1784 N (%) |
|---|---|
| Pap-smear appointment booked | 686 (38.4) |
| HPV self-test was sent | 445 (24.9) |
| Stated that they had recently provided a Pap smear | 93 (5.2) |
| Rejected offer for Pap smear booking or HPV self-test but were encouraged to attend drop in for Pap smear | 297 (16.6) |
| Stated that they had been hysterectomised | 104 (5.8) |
| Wanted to be omitted from the cervical cancer screening programme | 148 (8.3) |
| Missing information, excluding information about Pap smear or HPV testing | 11 (0.6) |

**Table 3. Preferred test in specific age groups.**

| Age group | Pap smear N = 686 N (%) | HPV self-test N = 445 N (%) | p-value |
|---|---|---|---|
| **30–39** | 138 (20.1) | 65 (14.6) | P<0.001 |
| **40–49** | 231 (33.7) | 124 (27.9) | |
| **50–65** | 314 (45.7) | 255 (57.3) | |
| **Missing**\* | 3 (0.5) | 1 (0.2) | |

\* Not included in the chi2 test

About two thirds (63.4%,1131/1784) of the women who answered the phone wanted to participate, either by booking a visit for a Pap smear (38.5% 686/1784) or by being sent an HPV self-test (24.9% 445/1784) (p<0.001)(Table 2). The preferred sample (Pap smear vs HPV self-test) in relation to age group is shown in Table 3.

The women in the 50–65 years age group preferred an HPV self-test while the women in the 30–39 age group preferred the Pap smear (p<0.001). Of the 686 women that were booked for a Pap smear, 58.9% (N = 404) came to the visit, and of the 445 women who had chosen an HPV self-test, 36.2% (N = 161) returned it (p<0.001) (Fig 1). Reasons for not having attended the cervical cancer screening programme were reported by 1221 women. The most common reasons were that they did not prioritise it or found insufficient accessibility (43.5% N = 531), felt fear/anxiety or resistance towards the examination (20.4% N = 249), had insufficient knowledge (4.4% N = 54), felt healthy (4.0% N = 49) and other reasons that the midwife was not able to specify (27.7% N = 338).Women who chose an HPV self-test over a Pap smear more often reported fear/anxiety 32.8% vs 11.8%, as the reason for not being compliant with the cervical cancer screening programme (p<0.001).

In the *control group*, 250 Pap smears were collected between the 8th of March 2016 and six months after that (7.0%, N = 250/ 3538) (Fig 1, Table 1).

## Comparison of participation between the study groups

A comparison between the three study groups regarding index samples is shown in Table 1. A significant difference is shown between all groups both in intention to treat and by protocol calculations. The telephone group had 25.5% index samples compared to 34.1% and 7.0% in the HPV self-test group and control group respectively (p<0.001) (intention to treat). In the by protocol analysis including women that answered the phone call the participation rates were 31.7% vs 26.1% and 7.0% (p<0.001) respectively (Table 1). The corresponding results in the by protocol analysis including women that did not answer the phone call were 19.7% vs 26.1% and 7.0% (p< 0.001) (Table 1).

The women that had a telephone call before the HPV self-test was sent returned it (36.1% N = 161/445) more often compared with the women that only had the HPV self-test sent by post with or without a reminder letter (16.6% N = 498/3002) (p<0.001) (9.7% N = 290/3002) (p<0.001).

## Attendance at follow-up, and abnormal test results

In the HPV self-test group, 13.1% of the tests (N = 103/786) were positive for HPV, and 84.5% of the women (N = 87/103) came to a follow-up visit. In the telephone group, 15.0% of the women who returned an HPV self-test were positive for HPV (N = 24/160), and 70.1% (N = 17/24) came for a follow-up. In total, 81.5% (N = 104/128) of women with positive HPV

**Table 4. Results of the positive HPV self-tests.**

| | Telephone group With HPV self-test N = 24 N (%) | HPV self-test group N = 103 N (%) | Total N = 127 N (%) |
|---|---|---|---|
| **HPV 16** | 4 (16.7) | 9 (8.7) | 13 (10.2) |
| **HPV18** | 1 (4.2) | 3 (2.9) | 4 (3.1) |
| **Non HPV16/18** | 17 (70.8) | 79 (76.7) | 96 (75.6) |
| **Other HPV combinations*** | 2 (8.3) | 12 (11.7) | 14 (11.0) |

*For example non HPV16/18 and HPV16

(13.4% 127/946) came for a follow-up. The distribution of the positive HPV results is shown in Table 4.

The cytology diagnoses from the index Pap smears in the telephone group and the control group are shown in Table 5.

Of the 642 index Pap smears collected in the telephone group, 7.8% (N = 50/641) had dysplasia (Table 5). All of these women came for a follow-up visit (N = 50/50). There were significant differences between the groups, where attending a follow-up after Pap smear (50/50; 100%) was significantly different from follow-up of positive HPV in the telephone group (17/24; 70.8%; p <0.000) and in the HPV self-test group (87/103; 84.5%; p = 0.003 respectively). In total, the attendance at the recommended follow-up was 87% (154/177).

For comparison, a group of women (N = 44938) that had completed the two previous cervical cancer screening procedures were included in the analysis. Non-attendees had a three or eight times higher risk of having a HSIL or suspected SCC in cytology respectively compared to those women (OR 3.3; CI 95% 1.9–5.2, OR 8.6; CI 95% 1.6–30) (Table 5). Results from the follow-up Pap smear of the women with positive HPV are not included in this analysis, since they were a selected group due to known HPV, and thereby at increased risk of dysplasia.

Biopsies from the cervix were taken from 66.3% (N = 69/104) of non-attendees with positive HPV and from 72% (N = 36/50) with cervical dysplasia in their Pap smears. In histopathology a total of ten SCCs and one adenocarcinoma were found in the non-attendee group with a study intervention compared to one SCC in the non-attendee control group (p = 0.02, OR 8.1 CI 95% 1.2–350). The distribution among the non-attendees concerning histopathology diagnosis was similar (Table 6).

**Table 5. Cytology results (intention to treat).**

| | Telephone group Pap smear identified within 6 months after invitation Letter was sent (N) | Control group Pap smear Identified 6 months after 8 the March (N) | SummaryNon-attendees N (%) | Women participating in the cervical screening program in the last two rounds N (%) | OR (95% conf. interval) |
|---|---|---|---|---|---|
| **Pap smears in total** | 642 | 250 | 892 | 44938 | |
| **Benign** | 591 | 236 | 827 (93.7) | 42771 (95.2) | Ref 1 |
| **LSIL*** | 26 | 10 | 36 (4.0) | 1732 (3.9) | 1.1 (0.7–1.5) |
| **HSIL**** | 18 | 3 | 21 (2.6) | 330 (0.73) | 3.3 (1.9–5.2) |
| **AIS***** | 4 | | 4 (0.4) | 80 (0.18) | 2.6 (0.6–6.9) |
| **Suspected SCC** | 2 | 1 | 3 (0.3) | 18 (0.04) | 8.6 (1.6–30) |
| **Missing** | 1 | | 1 (0.1) | 7 (0.02) | |

*Including ASCUS

** including ASC-H

*** Adenocarcinoma in situ

**Table 6. Histopathological results among non-attendees\* (intention to treat).**

| | Telephone group Pap Smear N = 36 | Telephone Group HPV self-Sampling test N = 12 | HPV self-sampling–test group N = 57 | Summary Non-attendees with intervention N = 105 | Control group N = 14 | OR (95% conf. interval) |
|---|---|---|---|---|---|---|
| Benign | 13 | 7 | 14 | 34 | 3 | Ref 1 |
| LSIL | 8 | 2 | 24 | 34 | 5 | 0.6 (0.08–3.4) |
| HSIL | 9 | 2 | 13 | 24 | 5 | 0.4 (0.06–2.4) |
| AIS\*\* | 1 | | | 1 | | |
| SCC | 5 | | 5 | 10 | 1 | 0.9 (0.06–51) |
| Adeno carcinoma | | | 1 | 1 | | |
| Unspecified | | 1 | | 1 | | |

\*Until 31 December 2017

\*\* Adenocarcinoma in situ

## Discussion

In this randomised population-based controlled trial we found that the best intervention to get as many non-attendees as possible to participate was to send an HPV self-test along with an invitation letter. We also found that almost all non-attendees came to a follow-up visit if the index sample was abnormal. More than one third of the women in the HPV self-test group participated, compared to one quarter of the women in the telephone group and less than one tenth in the control group. This indicates that the intervention of sending an HPV self-test to non-attendees is more effective as a method for obtaining participation compared to the two other approaches since some of these women also attended for a Pap smear after receiving the letter with the study invitation. Similar studies have shown that HPV self-tests and telephone calls by midwives both give significantly higher response rates than a control group, and therefore our result is in alignment with these studies [7,11,12,15,16]. Another recently published study has shown that the response rate after receiving an HPV self-test was slightly lower compared to our results (13.2%) but increased with age [17]. In contrast, to our study no reminder letter was sent but this could only explain some of the differences.

On the other hand, if the interventions are compared by protocol, a telephone call from a midwife with an answer from the women is the most effective method of achieving participation. But since only two thirds of the women answered the phone call this intervention faces major challenges, which are exacerbated due to the difficulties in finding correct phone numbers. If no answers is received the intervention of sending an HPV self-test is more effective for achieving participation.

Of the non-attendees in the telephone group that answered a telephone call from a midwife and wanted a booked appointment for a Pap smear, almost two thirds had a Pap smear taken, while one third of the women that wanted an HPV self-test returned the tests. Furthermore, women who answered the telephone call and chose an HPV self-test were more likely to return it compared to women in the HPV self-test group, no matter whether a reminder letter was sent or not. Since only about 10 percent of those who returned the HPV tests did so after receiving a reminder letter, this does not seem effective in a clinical context. Overall, this indicates that personal contact with a midwife and the opportunity to choose the intervention on one's own gives additional motivation to participate in the cervical cancer screening programme compared to just receiving an invitation letter.

The women answering the telephone call from a midwife were slightly older than the women that were not reached, and the majority answered on the first call. In total, more than

two thirds of women answered the phone calls after three attempts. This is almost at the same level as a previous study where 64% of called non-attendees were reached in four attempts. In that study, up to 10 attempts were made and finally they reached 80% [7]. However, the involved midwives argued that two or three calls would have been enough since the answer rate fell at each attempt [18]. The ultimate number of attempts to reach a non-attendee in relation to cost-effectiveness needs further investigation. In our study, a maximum of three telephone calls were made, mainly during the evenings between 5–8 pm. Broberg et al. called women between 8 am to 5 pm and reached a similar answer rate as ours after four attempts. In their study they suggested that the answer rate might have been higher if the phone calls had been made during the evening, which our study also indicates [7].

The majority of women answering the telephone call wanted to participate in the study and more often chose a Pap smear in favour of the HPV self-test. The reason for this could be that they were familiar with this test. These women also left an index sample more often than those who chose an HPV self-test. The women that wanted an HPV self-test were older and more often gave the reason of anxiety and fear for not attending the cervical cancer screening programme compared to the women that chose a Pap smear. It seems that older women, which previously declined to provide a Pap smear due to anxiety or fear now find an opportunity in the HPV self-test that is easier to adhere to. To our knowledge this has not been investigated before.

The main reason for not attending the cervical cancer screening programme was insufficient accessibility and lack of time. This finding is yet another wake-up call to all the outpatient clinics that are responsible for offering appointments to women for a Pap smear to be taken. Other studies show similar reasons for not participating [8, 9].

One limitation of this study is that we had to exclude women between the date of randomisation and the date of the invitation letter because the women no longer fulfilled the criteria of being non-attendees. In the telephone group one reason for exclusion was lack of a telephone number which was not the case in the HPV self-test group. We were not able to control for the effect of the routine yearly invitation on participation. However, this invitation applied equal to all groups of non-attendees and should therefore not affect the result. Our study has several strengths, such as being randomised, population-based, and carried out in an area with an already well-functioning cervical cancer screening programme. It could be assumed that the results can be generalised to other arears with a well-organized screening program for cervical cancer.

Loss of follow-up among non-attendees with an abnormal index sample must be minimised in order to maximise the desired benefit from extra interventions. The attendance at recommended follow-up visits after an abnormal index sample was almost 90 percent. Our results are comparable to the results achieved in other studies where about 90% of non-attendees with abnormal tests came to the first follow-up visit [7,15,19]. In a Norwegian trial even more (94.1%) women who tested positive for HPV by HPV self-test attended follow-up [16]. In our study, 84.5% of the HPV self-test group attended follow-up visits; however only 70.1% of the women who chose an HPV self-sampling test in the telephone group attended follow-up. One reason for the discrepancy between the follow-up rate of a positive HPV in the telephone group and HPV self-test group might be that women in the telephone group chose an HPV test rather than a Pap smear, and were still anxious about the gynaecological examination. The women in the telephone group that presented with an abnormal Pap smear had a follow-up attendance rate of 100%. This is higher than in another study where 87% of women with atypical Pap smears attended follow-ups after a telephone call with a midwife [7]. Interventions to increase participation among non-attendees are therefore meaningful because a majority (87–94%) of former non-attendees take part in recommended treatments and investigations, which

is also found in other studies [7,15,16, 19]. The prevalence of non-attendees found in this population (11.3%) is higher than in a similar Swedish study (7%), probably due to the different screening coverage [7]. About one in 20 of the non-attendees that were called stated that they had undergone a total hysterectomy, which is a valid reason not to attend the screening. The number of hysterectomies found in non-attendees in our study is lower than in a previous study where 12.6% of called women reported a hysterectomy [7]. Unregistered hysterectomies may cause a bias concerning participation rates in cervical cancer screening. If the number of unregistered women that have undergone a hysterectomy in the telephone arm is representative of all non-attendees in Sweden, the number of "real" non-attendees in the Östergötland region and other Swedish regions is lower than registered today.

In total, 13.4% of the non-attendees were positive for HPV. This result was higher compared to other studies showing an HPV positivity in the range of 8.3–10.3% among non-attendees compared to the approximately 8% HPV prevalence in the background population [19–21]. Only three HPV self-tests in our study could not be analysed (0.3%), which is in alignment with results from other studies [19, 20]. The non-attendees in our study had a three or eight times higher risk of having an HSIL or suspected SCC in cytology compared to the women in the attendee group. In the histopathology examinations ten SCCs and one adenocarcinoma were found among the non-attendees with a study intervention (the Telephone group and HPV self-test group) compared to one SCC in the control group of non-attendees. This indicates that non-attendees have a higher risk of developing cervical cancer compared to the background population and that an intervention makes a significant difference since cancers may be detected at an earlier stage of the disease. These results are in alignment with several other studies [7, 11,12,15,16,22].

## Conclusion

Our randomised population-based study suggests that the best method to get as many non-attendees as possible to participate is to send an HPV self-test along with an invitation letter. However, more women prefer to provide a Pap smear when given the opportunity to choose between a Pap smear and an HPV self-test. Women who choose an HPV self-test are older and give anxiety/ fear as a reason to decline participation. They are also less likely to attend the follow-up if the HPV is positive. As expected, more cytology diagnoses of HSIL and suspected SCC were found among the non-attendees participating in this study compared with a screened attendee group of women. Since about 90% of women with an abnormal index sample attended follow-up we consider this high enough to indicate that interventions to increase the participation rate among non-attendees are meaningful.

## Supporting information

**S1 Checklist. CONSORT 2010 checklist of information to include when reporting a randomised trial**[*]**.**
(DOC)

**S1 Data. Forskningsplan.**
(DOCX)

**S2 Data. Komplettering till ansökan om.**
(DOCX)

**S3 Data. Additional information to the first application.**
(DOCX)

**S4 Data. Project plan.**
(DOCX)

## Acknowledgments

To all midwives participating in the study especially Åsa Råsbrink and Maria Åkerlund. Maria Gruber for handling all the HPV self-tests. Administrator Therese Andersson- Mats Fredrikson for statistical help.

## Author Contributions

**Conceptualization:** Caroline Lilliecreutz.

**Data curation:** Hanna Karlsson.

**Formal analysis:** Caroline Lilliecreutz, Hanna Karlsson, Anna-Clara Spetz Holm.

**Funding acquisition:** Caroline Lilliecreutz.

**Investigation:** Caroline Lilliecreutz.

**Methodology:** Caroline Lilliecreutz, Anna-Clara Spetz Holm.

**Project administration:** Caroline Lilliecreutz.

**Resources:** Caroline Lilliecreutz.

**Supervision:** Caroline Lilliecreutz.

**Validation:** Caroline Lilliecreutz, Anna-Clara Spetz Holm.

**Visualization:** Caroline Lilliecreutz, Anna-Clara Spetz Holm.

**Writing – original draft:** Caroline Lilliecreutz, Hanna Karlsson, Anna-Clara Spetz Holm.

**Writing – review & editing:** Caroline Lilliecreutz, Anna-Clara Spetz Holm.

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
