## [Decision Letter · Decision Letter 0]

16 Mar 2020

PONE-D-20-00773

Participation in interventions and recommended follow-up for non-attendees in cervical cancer screening -taking the women’s own preferred test method into account - a Swedish randomised controlled trial

PLOS ONE

Dear Dr Lilliecreutz,

Thank you for submitting your manuscript to PLOS ONE. After careful consideration, we feel that it has merit but does not fully meet PLOS ONE’s publication criteria as it currently stands. Therefore, we invite you to submit a revised version of the manuscript that addresses the points raised during the review process.

We would appreciate receiving your revised manuscript by Apr 30 2020 11:59PM. To enhance the reproducibility of your results, we recommend that if applicable you deposit your laboratory protocols in protocols.io, where a protocol can be assigned its own identifier (DOI) such that it can be cited independently in the future. For instructions see: http://journals.plos.org/plosone/s/submission-guidelines#loc-laboratory-protocols

We look forward to receiving your revised manuscript.

Kind regards,

Joakim Dillner, M.D.

Academic Editor

PLOS ONE

Additional Editor Comments (if provided):

Dear Dr. Lilliecreutz,

The reviews are now completed and I would like to invite you to resubmit a revised version of the paper that takes the reviewers´commenst into account.

In addition:

The paper describes that the participants were enrolled before an IRB approval was obtained. This could have been a cause for a rejection without reviews, but I decided to give you the opportunity to explain. Please include this both in the accompanying letter when you resubmit the paper and in the revised manuscript itself.

From the comments, I am concerned about that it is not clear if the outcome is measured by cytology or histopathology, that it is not clear what the difference is between per protocol and intention to treat and how these 2 analyses could even have different conclusions. Please give exact counts (not only percentages) already in the abstract. Finally, please have a native English speaker revsie the paper before you resubmit.

I look forward to receive the revised manuscript,

Best Regards Joakim Dillner, Editor

Journal Requirements:

2. We note that you have reported significance probabilities of 0 in places. Since p=0 is not strictly possible, please correct this to a more appropriate limit, eg 'p<0.0001'.

4. Please provide additional details regarding participant consent. In the ethics statement in the Methods and online submission information, please ensure that you have specified whether consent was written or verbal/oral. If consent was verbal/oral, please specify how verbal/oral consent was recorded.

5. Please include in the Methods the rationale behind patients being enrolled prior to obtaining IRB approval.

6. We note that you have indicated that data from this study are available upon request. PLOS only allows data to be available upon request if there are legal or ethical restrictions on sharing data publicly. For information on unacceptable data access restrictions, please see http://journals.plos.org/plosone/s/data-availability#loc-unacceptable-data-access-restrictions.

Reviewers' comments:

Reviewer's Responses to Questions

**Comments to the Author**

1. Is the manuscript technically sound, and do the data support the conclusions?

Reviewer #1: Partly

Reviewer #2: Partly

2. Has the statistical analysis been performed appropriately and rigorously? 

Reviewer #1: Yes

Reviewer #2: Yes

3. Have the authors made all data underlying the findings in their manuscript fully available?

Reviewer #1: No

Reviewer #2: Yes

4. Is the manuscript presented in an intelligible fashion and written in standard English?

Reviewer #1: No

Reviewer #2: Yes

5. Review Comments to the Author

Reviewer #1: Long-term non-attenders to cervical screening have a higher risk of cervical cancer; therefore, determining effective ways to reach women who do not respond to routine invitations is important for the effectiveness of the screening program. This paper examines the effect on participation, follow-up, and disease detection of two active interventions involving self-sampling with the routine practice of sending a reminder invitation. The authors have also compared to a population that routinely attends cervical screening. While the premise is important, there are several aspects of the paper that need clarification and structuring to better highlight the study protocol and the results.

1) The materials and methods section needs to be reorganized for clarity – statements regarding registration of the trial can be moved to the end; information on the extraction of the comparison group can be included under the “attended group” heading along with further details on this population and reasoning on the selection criteria; a description of the ITT and per protocol population definitions should be included in the analysis section.

2) In the second paragraph of the methods, the authors state that all three groups received the yearly invitations. Is it possible to separate the effects of the intervention and the routine yearly invitations? Did they occur simultaneously or sequentially?

3) More details on the interventions would be helpful for reproducibility and understanding effects. What information did women receive about the intervention procedure? More details on the HPV self-sampling are needed – what test was used, were instructions included?

4) The authors have explored a series of outcomes related to the screening process but what is most interesting is the comparison between the interventions at each stage. That said, it would be helpful for the reader if the results were organized by outcome rather than a mix of intervention arm and outcome. As it stands, the results are somewhat hard to follow. The first stage in the results are framed as “obtaining an index sample”, would it be fair to say that this reflects “participation” and that the interventions are differently effective in increasing participation as compared to the control group? This interpretation might be more understandable to the reader and corresponds better to the screening process reflected in the choice of the other outcomes.

5) Perhaps it should be noted again that Table 5 reflects cytological results, and therefore *suspected* cancer, correct? If I’ve understood it correctly, these are not histopathologically confirmed cancer whereas Table 6 reflects results that are histopathologically confirmed. Furthermore, the description of Table 6 needs to be edited. The OR mentioned in line 295 is from Table 5 not Table 6 (this matters for the results summarized in the abstract as well). This needs to be clarified in the final paragraph of the Discussion (starting at Line 382).

6) While the authors have detected more underlying disease (and suspected cases of cancer) in the intervention arms as compared to the general population that attends screening regularly, the authors have made this conclusion based on cytology results which precludes information on staging. Therefore, statements on the downstaging effect of the interventions should be made with more caution at the end of the Discussion.

Smaller edits

1) Ref 4 - HPV16/18 cause 70% of all cervical cancer, not just SCC.

2) Please read through the text again – there are several typos that make the text less readable.

3) I may have missed something, but the dates for the interventions described in the second paragraph of the materials and methods section don’t line up with the dates described for each intervention separately.

4) I suggest using “abnormality” or “positive screening result” to describe the index tests that needed further follow-up rather than “pathology”. Also, I suggest replacing “PAD” in Table 6 with “histopathological results”. Furthermore, it can be noted that HSIL/LSIL terminology is used for both cytology and histology in this context to clarify for the reader.

Reviewer #2: Lilliecreutz et al., studied outcome of two intervention arms and one control group among non-attendees in cervical cancer screening.

They randomized 10.500 women to three groups among those that had not participated in > 6 years. Among the group of women who were randomized to get a HPV self-sample kit, 34.1% sent in an index sample (either HPV-sample or a pap-smear). Corresponding figures were 25.5% and 7.0% (cytology) for women that was invited by telephone and by routine reminder by post, respectively. The majority of the women 63.4% who answered the telephone wanted to have a pap smear (38.5%) or to be sent a HPV self- sampling kit (24.9%) (p<0.001). Women who chose an HPV self-test were older and gave anxiety/ fear as a reason to decline participation, and they were also less likely to participate in the follow-up if HPV-positive compared to the women who chose a Pap smear. In the by protocol analysis the telephone group (among those who answered) was the most effective regarding collection of an index sample compared to the HPV self-sampling test and control group with 31,7% vs 26,1% and 7,0% ( p<0.001) respectively in each group. The attendance to the recommended follow-up in total was 87%. Non-attendees had a three respectively eight times higher risk of having a HSIL or SCC compared to women screen as recommended (OR 3.3 CI 95% 1.9-5.2, OR 8.6 CI 1.6-30). A total of ten SCC and one adenocarcinoma were found in the non-attendee group with a study intervention compare to one SCC in the non attendee group without any study intervention ( p=0.02, OR 8.1 CI 95% 1.2-350).

They concluded that the best strategy to get more non-attendees to participate is to send a HPV-self sample kit.

The study is interesting due to its size of invited population based women and the engagement of 24 midwifes in order to perform the telephone arm.

Comments

The manuscript will benefit from improvements as mentioned below.

1. Abstract. Please clarify which result that was used to conclude that the best intervention method to get as many non-attendees as possible to leave a sample is to send a HPV self sampling test together with an invitation letter? In contrast (line 344) in the discussion you favour personal contact with a midwife with an option for the woman to choose sampling method.

2. Please describe that “intention to treat statistical analysis” was performed in both abstract and materials and methods.

3. Please clarify “by protocol” in the Material and Method section.

4. Line 179 ASC-H? Please change to ” that cannot exclude a high-grade squamous intraepithelial lesion”

5. Line 368. Please clarify the term “majority”?

6. Line 337. Please consider a recent study which also observed increased return rates of self-samples by increasing age (PMID: 32082413).

7. Table 1. Please give explanations (footnotes) of “Intention to treat”, “Index sample”, “By protocol” and add headings in the two left bottom squares.

6. PLOS authors have the option to publish the peer review history of their article (what does this mean?). If published, this will include your full peer review and any attached files.

Reviewer #1: No

Reviewer #2: No

---

## [Author Response · Author response to Decision Letter 0]

22 Apr 2020

Dear Academic Editor, Professor Dillner

Thank you for valuable comments on our manuscript “Participation in interventions and recommended follow-up for non-attendees in cervical cancer screening -taking the women’s own preferred test method into account - a Swedish randomised controlled trial”.

We have tried to change the manuscript in accordance to the suggestions made by the reviewers and yourself, please see below. 

If you still think there are modifications that need to be done in order to make the manuscript publishable, please do not hesitate to contact us. But our sincere hope is that you will find the manuscript satisfactory for publication.

The paper describes that the participants were enrolled before an IRB approval was obtained. This could have been a cause for a rejection without reviews, but I decided to give you the opportunity to explain. Please include this both in the accompanying letter when you resubmit the paper and in the revised manuscript itself.

The first decision from IRB came 27 th of January 2016 with a request that we needed to change some of the content in the information to the study population. The final approval from the IRB came 12th of April 2016 after the time of the randomization took place (March 2016) but before the invitations were send to the patients. Revisions and clarifications has been made in the manuscript.

From the comments, I am concerned about that it is not clear if the outcome is measured by cytology or histopathology, Thank you for your comment. We have clarified the outcomes concerning cytology and histopathology throughout the manuscript.

that it is not clear what the difference is between per protocol and intention to treat and

We have added a definition in the statistic section and hope that it will be to your satisfaction

how these 2 analyses could even have different conclusions. 

Since we only included women who answered the phonecall in the by protocol analyses, the studygroup is much smaller than in the intention to treat group, which contribute to the differences in the conclusions. We have added measures in the results-section and clarified this in the discussion section. We also made changes in table 1.

Please give exact counts (not only percentages) already in the abstract. 

Thank you for the comment. This has been added.

Finally, please have a native English speaker revise the paper before you resubmit.

We have sent it back to Anchor English, - Proofreading Services https://www.anchorenglish.com who has revised the manuscript once again

 Journal Requirements:

We note that you have reported significance probabilities of 0 in places. Since p=0 is not strictly possible, please correct this to a more appropriate limit, eg 'p<0.0001'. Changes has been done in the manuscript

We suggest you thoroughly copyedit your manuscript for language usage, spelling, and grammar. If you do not know anyone who can help you do this, you may wish to consider employing a professional scientific editing service. 

The manuscript has been copyedit for language, spelling and grammar by Anchor English - Proofreading Services https://www.anchorenglish.com

Caroline Lilliecreutz and Anna-Clara Spetz-Holm has been editing the manuscript. 

Please provide additional details regarding participant consent. In the ethics statement in the Methods and online submission information, please ensure that you have specified whether consent was written or verbal/oral. If consent was verbal/oral, please specify how verbal/oral consent was recorded.

The consent was giving verbal and confirmed by taking a sample (pap smear or HPV self-test) after study information was given. Revisions had been made.

Please include in the Methods the rationale behind patients being enrolled prior to obtaining IRB approval.

This has been clarified. Thank you.

6. We note that you have indicated that data from this study are available upon request. PLOS only allows data to be available upon request if there are legal or ethical restrictions on sharing data publicly. For information on unacceptable data access restrictions, please see http://journals.plos.org/plosone/s/data-availability#loc-unacceptable-data-access-restrictions.

Data is available on request for any researchers to allow replication of results provided all ethical and legal requirements are met. Data can be requested through Forum Östergötland, Linköping University, with contact person Mats Fredrikson Mats mats.fredrikson@liu.se. Registries used for this study include the Swedish National Cervical Cancer Registry and A local database (HPV data). Address to the ethics committee: www. Etikprovningsmyndigheten .se. E- mail; registrator@etikprovning.se.

See comment above

Please include captions for your Supporting Information files at the end of your manuscript, and update any in-text citations to match accordingly. Please see our Supporting Information guidelines for more information: http://journals.plos.org/plosone/s/supporting-information.

We do not have any supporting information files to publish. If we have misunderstood the instruction, please let us know.

Review Comments to the Author

Reviewer #1: Long-term non-attenders to cervical screening have a higher risk of cervical cancer; therefore, determining effective ways to reach women who do not respond to routine invitations is important for the effectiveness of the screening program. This paper examines the effect on participation, follow-up, and disease detection of two active interventions involving self-sampling with the routine practice of sending a reminder invitation. The authors have also compared to a population that routinely attends cervical screening. While the premise is important, there are several aspects of the paper that need clarification and structuring to better highlight the study protocol and the results.

1) The materials and methods section needs to be reorganized for clarity

• Statements regarding registration of the trial can be moved to the end. Revisions has been made.

• Information on the extraction of the comparison group can be included under the “attended group” heading along with further details on this population reasoning on the selection criteria;

Thank you for the comment. The description of the control group is described earlier in the methods section. We added more information of the attendee-group earlier in the methods section. Due to these changes we omitted sections “control group” and “attended group”. 

• a description of the ITT and per protocol population definitions should be included in the analysis section.

Thank you, we have added information about this in the statistics section under Material and Methods

2) In the second paragraph of the methods, the authors state that all three groups received the yearly invitations. Is it possible to separate the effects of the intervention and the routine yearly invitations? Did they occur simultaneously or sequentially? Since all three groups received the routine yearly invitations the effect of this invitation exists in all groups. The study interventions are in addition. A clarification has been made in the discussion section and we also added information in the Methods section.

3) More details on the interventions would be helpful for reproducibility and understanding effects. What information did women receive about the intervention procedure? More details on the HPV self-sampling are needed – what test was used, were instructions included?

More details about the HPV self-sampling has been added- se Method section.

4) The authors have explored a series of outcomes related to the screening process but what is most interesting is the comparison between the interventions at each stage. That said, it would be helpful for the reader if the results were organized by outcome rather than a mix of intervention arm and outcome. As it stands, the results are somewhat hard to follow. The first stage in the results are framed as “obtaining an index sample”, would it be fair to say that this reflects “participation” and that the interventions are differently effective in increasing participation as compared to the control group? This interpretation might be more understandable to the reader and corresponds better to the screening process reflected in the choice of the other outcomes.

Thank you, we understand the complexity of our results. We have made some changes in the results section and changed the headlines to better clarify this. We hope this makes it clearer. We choose to keep “index sample” in some instances we found appropriate, and changed to “participation” when suitable. 

5) Perhaps it should be noted again that Table 5 reflects cytological results, and therefore *suspected* cancer, correct? This has been added. If I’ve understood it correctly, these are not histopathologically confirmed cancer whereas Yes that’s correct

Table 6 reflects results that are histopathologically confirmed. 

Yes that is correct

Furthermore, the description of Table 6 needs to be edited. 

We have changed the description to this Table

The OR mentioned in line 295 is from Table 5 not Table 6 (this matters for the results summarized in the abstract as well). This needs to be clarified in the final paragraph of the Discussion (starting at Line 382).

The OR is from Table 5 and we have clarified that in the results section concerning cytology (suspected SCC) and not histopathology. We have also added clarifications concerning this in the abstract and discussion.

6) While the authors have detected more underlying disease (and suspected cases of cancer) in the intervention arms as compared to the general population that attends screening regularly, the authors have made this conclusion based on cytology results which precludes information on staging. Therefore, statements on the downstaging effect of the interventions should be made with more caution at the end of the Discussion.

Thank you, sorry for confusion in the manuscript, we have made changes in the end of the discussion section to clarify this. Since the results from our study groups reflects histopathology we think that the conclusion that this indicates that the non-attendees has a higher risk of SCC than attendees is fair enough. 

Smaller edits

1) Ref 4 - HPV16/18 cause 70% of all cervical cancer, not just SCC. Thank you. Revision has been made.

2) Please read through the text again – there are several typos that make the text less readable.

We have once again sent it back to Anchor English, - Proofreading Services https://www.anchorenglish.com who has revised the manuscript

3) I may have missed something, but the dates for the interventions described in the second paragraph of the materials and methods section don’t line up with the dates described for each intervention separately.

Thank you, we have changed, however, the distribution of the invitation letters differed a little between the telephone group and the HPV self-test group.

4) I suggest using “abnormality” or “positive screening result” to describe the index tests that needed further follow-up rather than “pathology”. 

Changes has been made to meet this

Also, I suggest replacing “PAD” in Table 6 with “histopathological results”. 

Changes has been made

Furthermore, it can be noted that HSIL/LSIL terminology is used for both cytology and histology in this context to clarify for the reader.

Thank you for your comment, however vi have tried to clarify this throughout the manuscript.

Reviewer #2: Lilliecreutz et al., studied outcome of two intervention arms and one control group among non-attendees in cervical cancer screening.

They randomized 10.500 women to three groups among those that had not participated in > 6 years. Among the group of women who were randomized to get a HPV self-sample kit, 34.1% sent in an index sample (either HPV-sample or a pap-smear). Corresponding figures were 25.5% and 7.0% (cytology) for women that was invited by telephone and by routine reminder by post, respectively. The majority of the women 63.4% who answered the telephone wanted to have a pap smear (38.5%) or to be sent a HPV self- sampling kit (24.9%) (p<0.001). Women who chose an HPV self-test were older and gave anxiety/ fear as a reason to decline participation, and they were also less likely to participate in the follow-up if HPV-positive compared to the women who chose a Pap smear. In the by protocol analysis the telephone group (among those who answered) was the most effective regarding collection of an index sample compared to the HPV self-sampling test and control group with 31,7% vs 26,1% and 7,0% ( p<0.001) respectively in each group. The attendance to the recommended follow-up in total was 87%. Non-attendees had a three respectively eight times higher risk of having a HSIL or SCC compared to women screen as recommended (OR 3.3 CI 95% 1.9-5.2, OR 8.6 CI 1.6-30). A total of ten SCC and one adenocarcinoma were found in the non-attendee group with a study intervention compare to one SCC in the non attendee group without any study intervention ( p=0.02, OR 8.1 CI 95% 1.2-350).

They concluded that the best strategy to get more non-attendees to participate is to send a HPV-self sample kit.

The study is interesting due to its size of invited population based women and the engagement of 24 midwifes in order to perform the telephone arm.

Comments

The manuscript will benefit from improvements as mentioned below.

1. Abstract. 

Please clarify which result that was used to conclude that the best intervention method to get as many non-attendees as possible to leave a sample is to send a HPV self sampling test together with an invitation letter? 

Thank you for your comment. We have clarified this in the abstract and this is according to the intention to treat-analyse.

In contrast (line 344) in the discussion you favour personal contact with a midwife with an option for the woman to choose sampling method.

Thank you, this is according to by protocol analyse, we have clarified this in the discussion. 

2. Please describe that “intention to treat statistical analysis” was performed in both abstract and materials and methods.

Thank you, we have added information in the abstract as well as in the material and methods section.

3. Please clarify “by protocol” in the Material and Method section.

We have clarified this in the Methods section under the section Statistics

4. Line 179 ASC-H? Please change to ” that cannot exclude a high-grade squamous intraepithelial lesion” 

Thank you. Changes has been made.

5. Line 368. Please clarify the term “majority”? Clarifications has been made.

6. Line 337. Please consider a recent study which also observed increased return rates of self-samples by increasing age (PMID: 32082413). Thank you. Very interesting data. This reference has been added.

7. Table 1. Please give explanations (footnotes) of “Intention to treat”, “Index sample”, “By protocol” and add headings in the two left bottom squares. Thank you this has been added. Definitions is also included in the statistics section.

---

## [Decision Letter · Decision Letter 1]

11 Jun 2020

Participation in interventions and recommended follow-up for non-attendees in cervical cancer screening -taking the women´s own preferred test method into account

- a Swedish randomised controlled trial

PONE-D-20-00773R1

Dear Dr. Lilliecreutz,

We’re pleased to inform you that your manuscript has been judged scientifically suitable for publication and will be formally accepted for publication once it meets all outstanding technical requirements.

Kind regards,

Joakim Dillner, M.D.

Academic Editor

PLOS ONE

Additional Editor Comments (optional):

Reviewers' comments:

Reviewer's Responses to Questions

**Comments to the Author**

1. If the authors have adequately addressed your comments raised in a previous round of review and you feel that this manuscript is now acceptable for publication, you may indicate that here to bypass the “Comments to the Author” section, enter your conflict of interest statement in the “Confidential to Editor” section, and submit your "Accept" recommendation.

Reviewer #3: All comments have been addressed

2. Is the manuscript technically sound, and do the data support the conclusions?

Reviewer #3: Yes

3. Has the statistical analysis been performed appropriately and rigorously? 

Reviewer #3: Yes

4. Have the authors made all data underlying the findings in their manuscript fully available?

Reviewer #3: No

5. Is the manuscript presented in an intelligible fashion and written in standard English?

Reviewer #3: Yes

6. Review Comments to the Author

Reviewer #3: The revision of the paper addressed most of the raised comments. Also the methods section improved.

Data were analysed with descriptive statistics. -> The authors use hypothesis testing. However, hypothesis testing is no descriptive statistics (it is inferential statistics, see

https://www.mdpi.com/2504-4990/1/3/54

The rejection of the null- hypothesis was set to 0.05 -> The significance level alpha was set to 0.05

7. PLOS authors have the option to publish the peer review history of their article (what does this mean?). If published, this will include your full peer review and any attached files.

Reviewer #3: No

---

## [Editor Report · Acceptance letter]

16 Jun 2020

PONE-D-20-00773R1 

Participation in interventions and recommended follow-up for non-attendees in cervical cancer screening -taking the women’s own preferred test method into account
- a Swedish randomised controlled trial 

Dear Dr. Lilliecreutz:

I'm pleased to inform you that your manuscript has been deemed suitable for publication in PLOS ONE. Congratulations! Your manuscript is now with our production department. 

Kind regards, 

on behalf of

Dr. Joakim Dillner 

Academic Editor

PLOS ONE